# LLM-FP4: 4-Bit Floating-Point Quantized Transformers

**Shih-yang Liu[*1], Zechun Liu[*2], Xijie Huang[1], Pingcheng Dong[1], Kwang-Ting Cheng[1]**
[1]Hong Kong University of Science and Technology, [2]Meta Reality Labs
{sliuau, xhuangbs, pingcheng.dong}@connect.ust.hk
zechunliu@meta.com
timcheng@ust.hk

## Abstract

We propose LLM-FP4 for quantizing both weights and activations in large language models (LLMs) down to 4-bit floating-point values, in a post-training manner. Existing post-training quantization (PTQ) solutions are primarily integer-based and struggle with bit widths below 8 bits. Compared to integer quantization, floating-point (FP) quantization is more flexible and can better handle long-tail or bell-shaped distributions, and it has emerged as a default choice in many hardware platforms. One characteristic of FP quantization is that its performance largely depends on the choice of exponent bits and clipping range. In this regard, we construct a strong FP-PTQ baseline by searching for the optimal quantization parameters. Furthermore, we observe a high inter-channel variance and low intra-channel variance pattern in activation distributions, which adds activation quantization difficulty. We recognize this pattern to be consistent across a spectrum of transformer models designed for diverse tasks, such as LLMs, BERT, and Vision Transformer models. To tackle this, we propose per-channel activation quantization and show that these additional scaling factors can be reparameterized as exponential biases of weights, incurring a negligible cost. Our method, for the first time, can quantize both weights and activations in the LLaMA-13B to only 4-bit and achieves an average score of 63.1 on the common sense zero-shot reasoning tasks, which is only 5.8 lower than the full-precision model, significantly outperforming the previous state-of-the-art by 12.7 points. Code is available at: https://github.com/nbasyl/LLM-FP4.

## 1 Introduction

Since the introduction of transformer architecture (Vaswani et al., 2017), transformers have superseded recursive neural networks, emerging as the dominant architecture in numerous natural language processing (NLP) tasks (Kenton and

---

*These authors contributed equally to this work

Toutanova, 2019; Lewis et al., 2020). The transformative impact of the transformer has been further propelled by the emergence of models like GPT (Brown et al., 2020; OpenAI, 2023), catapulting the popularity of this architecture to new heights. Meanwhile, the versatility of transformers extends beyond NLP, encompassing diverse domains such as vision (Dosovitskiy et al.; Touvron et al., 2021), audio (Akbari et al., 2021), etc. This trend towards a unified architecture for different modalities represents a groundbreaking development within the realm of deep learning.

However, the advancements in transformer performance are accompanied by a corresponding increase in model size and computational costs (Kaplan et al., 2020). This poses significant challenges when attempting to leverage the full potential of transformer models in use cases where memory or computational resources are limited. Despite the extensive research and widespread adoption of transformers, the field of transformer compression remains relatively underexplored. To address this gap, our study focuses on the compression of transformers, especially through floating-point post-training quantization techniques.

Post-training quantization (PTQ) offers the advantages of simple to use with minimal fine-tuning requirements (Nagel et al., 2020; Cai et al., 2020). Existing PTQ solutions for transformers primarily focus on integer (INT) quantization (Liu et al., 2021; Yuan et al., 2022), which can be effective in certain scenarios but often break down when bit widths are below 8 bit. On the other hand, floating-point (FP) quantization has gained significant traction as a more flexible alternative, capable of better accommodating various activation and weight distributions. In fact, FP8 has emerged as the default choice in various hardware platforms, including the NVIDIA H100.

Different from integer (INT) quantization, a particular challenge in floating-point (FP) quantiza-

tion is how to select appropriate exponent bits and scale parameters. Improper parameter choices can lead to subpar or divergent quantization results. To tackle this challenge, we introduce a robust recipe for FP quantization, which leverage layer-wise reconstruction to jointly search for optimal exponent bits and maximum values. Compared to previous approaches that utilize gradient updates for exponent bits (Kuzmin et al., 2022), our search-based method proves to be more stable and consistently delivers desirable quantization results, which establishes a strong baseline for FP-PTQ.

Furthermore, our investigation uncovers an intriguing pattern of activation distributions in transformers, characterized by high inter-channel variance and low intra-channel variance. Similar patterns are also observed in previous works (Xiao et al., 2022; Dettmers et al., 2022), while we argue that this pattern is inherent to transformer architectures and not limited to specific tasks, as we have observed consistent patterns not only in large language models but also in BERT model and even vision transformers. Motivated by these findings, we introduce a novel *pre-shifted exponent bias* for FP quantization of transformers. Concretely, we leverage the per-channel activation variance computed from calibration data and reparameterize these scales as the exponential bias of the corresponding FP quantized weight vectors. This approach effectively addresses the challenge posed by high inter-channel variance while incurring negligible computational cost.

In summary, we study floating-point post-training quantization (PTQ) for transformer architectures, and the contribution of this paper includes:

● We propose a search-based framework for determining the optimal exponent bias and maximal quantization value. This method outperforms existing techniques in terms of stability and performance, establishing a strong baseline for floating-point post-training quantization.

● We propose a novel technique, *pre-shifted exponent bias*, which effectively addresses the challenge of high inter-channel variance in the transformer with negligible computational overhead.

● Experimental results demonstrate that the proposed method yields the first usable FP4 weight and activation quantized LLaMA-13B model with mere 5.8-point degradation in zero-shot reasoning tasks against the full-precision model, reducing the gap by ∼70% compared to the previous SoTA.

● We further extend our method to BERT and vision transformers. It surpasses the previous best 4-bit quantized BERT by 7.8 points on GLUE dataset and achieves 31.4 points higher accuracy compared to the previous SoTA ViT quantization method for 4-bit DeiT-S on ImageNet dataset.

## 2 Related Works

### 2.1 Post-Training Quantization

Model quantization can be mainly categorized into quantization-aware training (QAT) and post-training quantization (PTQ), depending on whether it involves additional training for weight fine-tuning or not. Most PTQ studies are primarily focused on convolutional neural networks (CNNs) (Nagel et al., 2020; Li et al., 2021; Wu et al., 2020; Cai et al., 2020; Nagel et al., 2019). However, with the growing popularity of transformer-based models, only a limited number of works (Bondarenko et al., 2021; Yuan et al., 2022; Ding et al., 2022) have been conducted to realize PTQ on transformers. Moreover, the existing works primarily focus on visual transformer models and exhibit inferior performance when the bit width is below 8. Therefore, in this work, we delve into the challenges of the low-bit PTQ for language transformers.

### 2.2 Floating-Point Quantization

Floating-point (FP) quantization has emerged as a promising alternative to integer quantization due to its ability to handle long-tail distributions, and offers increased flexibility (Kuzmin et al., 2022). Additionally, modern GPUs such as H100 (Micikevicius et al., 2022) now support FP quantization. Nonetheless, minimal research has been conducted on FP quantization. Only (Kuzmin et al., 2022) proposes a general FP8 quantization scheme primarily for vision tasks, and (Zhang et al., 2023) adopts a mixture of FP and INT formats quantization for LLMs. In this work, we propose FPQ baseline as a general guideline for low-bit floating-point PTQ to compress language transformer models.

## 3 Preliminaries

### 3.1 Formulation of Floating-Point Variables

A standard floating-point number is represented as:

$$X_{\text{FP}} = (-1)^s 2^{p-b}(1 + \frac{d_1}{2} + \frac{d_2}{2^2} + ... + \frac{d_m}{2^m}) \quad (1)$$

where $s \in \{0, 1\}$ is the sign bit. $d_i \in \{0, 1\}$ is $i^{th}$ mantissa bit, $m$ denoted number of mantissa bits.

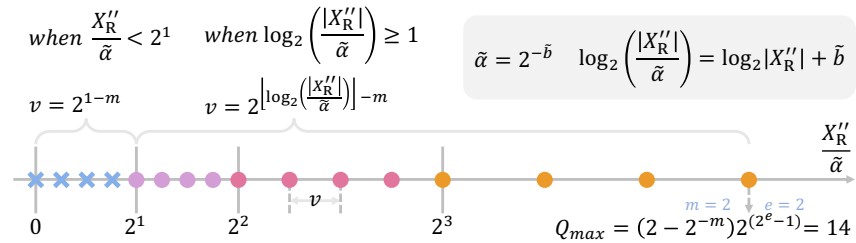

Figure 1: An illustration of floating-point (FP) quantization process using FP5 (E2M2) positive axis. The real-valued clipped $X_R''$ in Eq. 5 is rescaled by the real-valued scaling factor $\tilde{\alpha}$. Then, the quantization step-size $v$ is determined by the range $[2^p, 2^p + 1)$ in which $\frac{X_R''}{\tilde{\alpha}}$ falls (Eq. 9). Here, $p \in \{0, 1, ..., 2^{e-1}\}$ is the exponent bit value. Lastly, $X$ can be quantized to low-bit floating point values simply by $X_{\text{FP}} = \tilde{\alpha} \cdot v \cdot \left\lfloor \frac{X_R''}{\tilde{\alpha} \cdot v} \right\rceil$ (Eq. 8).

$p$ is an integer in $[0, 2^e - 1]$, and $e$ denotes number of exponent bits. $b$ is an integer exponent bias. A floating point with $j$ number exponent bits and $k$ mantissa bits is denoted as FP format EjMk.

### 3.2 Floating-Point Quantization Process

In integer quantization, the real-valued variable $X_R$ is quantized to an integer $X_{\text{INT}}$ with the following formula:

$$X_{\text{INT}} = \alpha \left\lfloor \text{Clip}\left( \frac{X_R}{\alpha}, Q_{min}, Q_{max} \right) \right\rceil \quad (2)$$

where $\lfloor \cdot \rceil$ is the rounding function. $X_R$ is the real-valued variable, $\alpha$ represents the full-precision scaling factor, and $Q_{min}$, $Q_{max}$ are the min/max value of the quantization range. Similarly, a real-valued variable $X_R$ can be converted to floating-point $X_{\text{FP}}$ in two steps.

(1) **Scale and clip.** In FP quantization, we also scale and clip the real-valued variable before quantization as:

$$X_R' = \text{Clip}(X_R, Q_{min}, Q_{max}) \quad (3)$$

where the min/max value range of signed floating-point quantization can be calculated from Eq.1:

$$Q_{max} = -Q_{min} = (2 - 2^{-m})2^{2^e - b - 1} \quad (4)$$

Here the integer exponent bias $b$ is another adjustable hyperparameter controlling $Q_{max}$ and $Q_{min}$, which has similar functionality as $\alpha$. Therefore, for simplicity, we reformulate Eq. 3 as:

$$X_R'' = \text{Clip}\left( X_R, \tilde{Q}_{min}, \tilde{Q}_{max} \right), \quad (5)$$

where

$$\begin{aligned}
\tilde{Q}_{max} &= \alpha Q_{max} = \alpha \cdot (2 - 2^{-m})2^{2^e - b - 1} \\
&= \alpha \cdot 2^{-b} \cdot (2 - 2^{-m})2^{2^e - 0 - 1} \\
&= 2^{-\tilde{b}} \cdot (2 - 2^{-m})2^{2^e - 0 - 1}
\end{aligned}$$
$$(6)$$

Note that we combine the tensor-wise real-valued scaling factor $\alpha$ with integer exponent bias $b$ to form a new scaling factor $\tilde{\alpha} = 2^{-\tilde{b}} = 2^{-b} \cdot \alpha$. Here $\tilde{b}$ denotes a relaxed tensor-wise real-valued exponent, and we can derive $\tilde{b}$ from the desired clipping value $\tilde{Q}_{max}$ from Eq. 6 as:

$$\tilde{b} = 2^e - \log_2 \tilde{Q}_{max} + \log_2(2 - 2^{-m}) - 1 \quad (7)$$

(2) **Compare and quantize.** Different from integer quantization, which simply utilizes the rounding function to convert the real-valued variables to quantized ones, in floating-point quantization, there is an additional step of comparing $X_R''$ with quantization levels and then quantize:

$$X_{\text{FP}} = \tilde{\alpha} \cdot v \cdot \left\lfloor \frac{X_R''}{\tilde{\alpha} \cdot v} \right\rceil \quad (8)$$

where $X_R''$ is clipped real-valued variable (Eq. 5), $\tilde{\alpha}$ is the tensor-wise floating-point scaling factor, and $v$ is an integer power of 2.

$$v = \begin{cases} 2^{\lfloor \log_2 |\mathbf{X}_R''| + \tilde{b} \rfloor - m} & \text{if } \lfloor \log_2 |\mathbf{X}_R''| + \tilde{b} \rfloor \geq 1 \\ 2^{1-m} & \text{otherwise} \end{cases} \quad (9)$$

Here we select the quantization level $v$ according to the magnitude of $\frac{X_R''}{\tilde{\alpha}}$, which equals to $X_R'' \cdot 2^{\tilde{b}}$. Then the floating-point quantized variables can be derived with Eq.8. The illustration of the quantization process is in Fig. 1, detailed explanation can also be found in (Micikevicius et al., 2022).

### 3.3 Floating-Point Matrix Multiplication

With the floating-point quantized variables, the matrix multiplication is formulated as:

$$\mathbf{O}_{out}^{i,k} = \mathbf{X}_{\text{FP}}^{i,:} \mathbf{W}_{\text{FP}}^{:,k} = \tilde{\alpha}_{\mathbf{x}} \tilde{\alpha}_{\mathbf{w}}^k \tilde{\mathbf{X}}_{\text{FP}}^{i,:} \tilde{\mathbf{W}}_{\text{FP}}^{:,k} \quad (10)$$

Here in per-tensor activation quantization and per-channel weight quantization, $\mathbf{X}_{\text{FP}}^{i,:}$ denotes $i^{th}$ row

in the activation matrix and $\mathbf{W}_{\text{FP}}^{:,k}$ denotes $k^{th}$ column in the weight matrix, such that each element $\mathbf{O}_{out}^{i,k}$ in the output matrix is computed by the product of two real-valued scalars $\tilde{\alpha}_{\mathbf{x}}$ and $\tilde{\alpha}_{\mathbf{W}}^k$ times the corresponding quantized activation and weight vectors. We depict all the possible quantization granularity options that support such efficient matrix multiplication in Appendix D.

## 4 Method

In this section, we begin by introducing our joint format and max value search, which establishes our strong baseline and already achieves state-of-the-art results at 8-bit and 6-bit quantization. Then we present an efficient *pre-shifted exponent bias* to tackle the catastrophic high inter-channel activation variance in transformer models and push the quantization limit to 4-bit.

### 4.1 Joint Format and Max Value Search

The objective of post-training quantization is to minimize the perturbation ($\delta\mathbf{X} = \mathbf{X}_{\text{FP}} - \mathbf{X}_{\text{R}}$) introduced by quantization to the pre-trained real-valued network:

$$\min \mathbb{E}[\mathcal{L}(\mathbf{X}_{\text{R}} + \delta\mathbf{X}) - \mathcal{L}(\mathbf{X}_{\text{R}})] \qquad (11)$$

In this study, we adopt the setting presented in (Choukroun et al., 2019; Wu et al., 2020), which assumes a positive correlation between the change in the intermediate output of the quantized model and Eq. 11. Therefore, minimizing the distance between the intermediate output of the quantized layer ($\hat{\mathbf{O}}$) and the output of the original layer ($\mathbf{O}$) leads to minimize Eq. 11. Hence, the objective loss metric is formulated as:

$$\min (\hat{\mathbf{O}} - \mathbf{O})^2 \qquad (12)$$

which is used to search for the optimal FP quantization function in the following proposed framework.

The challenges in FP quantization arise from its sensitivity to the quantization format and clipping range. Undesirable format selection will result in a catastrophic error rate. In addition, we observe that the optimal clipping range varies depending on the format used. Previous work (Kuzmin et al., 2022) on floating-point (FP) quantization-aware training (QAT) proposed to learn both the FP format and maximum value with gradients. However, we find this method suffers from over-fitting in PTQ, with accuracy being even worse than naïve MinMax

method, details can be found in Appendix E. Instead, we propose a search-based algorithm that jointly determines the optimal format and its associated clipping range to address this challenge.

The searching process is conducted layer by layer with the metric of minimizing Eq. 12. The output of matrix multiplication corresponding to each sub-module is denoted as $\mathbf{O} = \mathbf{XY}$, where $\mathbf{Y}$ can be either a weight tensor $\mathbf{W}$ or another activation tensor.

The search space of $q$-bit FP format includes all formats except for the format with an exponent bit equal to 0, as the quantization of the format with an exponent bit equal to 1 already degenerates to INT quantization. We search for the real-valued exponent bias $\tilde{b}$, which equals to the logarithm of the scaling factor. We initialize $\tilde{b}_{\mathbf{X}}$ and $\tilde{b}_{\mathbf{Y}}$ from Eq. 7 with $Q_{max}$ equals the maximum value of $|\mathbf{X}_{\text{R}}|$ and $|\mathbf{Y}_{\text{R}}|$, respectively. We then define the search space of $\tilde{b}_{\mathbf{X}}$ and $\tilde{b}_{\mathbf{Y}}$ by linearly dividing $[\gamma_1 \tilde{b}_{\mathbf{X}}^{init}, \gamma_2 \tilde{b}_{\mathbf{X}}^{init}]$ and $[\gamma_1 \tilde{b}_{\mathbf{Y}}^{init}, \gamma_2 \tilde{b}_{\mathbf{Y}}^{init}]$ into $k$ intervals, where $\gamma_1$ and $\gamma_2$ are empirically set to 0.01 and 1.2, and $k = 100$.

The search process is outlined in Alg.1. We search the quantization scheme in all the matrix multiplication layers in parallel following (Yuan et al., 2022; Bai et al., 2022). The algorithm can be divided into two parts. (1) Do forward propagation to store the intermediate raw output of each layer $l$. (2) Iteratively update the optimal format and biases for each layer for three rounds by minimizing the reconstruction metric (Eq. 12). We name this search-based framework as *Floating Point Quantization Baseline* (FPQ baseline), and it can already achieve state-of-the-art results on both 8-bit and 6-bit settings.

### 4.2 Pre-Shifted Exponent Bias

In transformer architectures, we observed an intriguing phenomenon of high inter-channel variance. As shown in Fig.2, the magnitudes of values within the same channel are close to each other but exhibit significant differences across different channels. This phenomenon is not only observed in language models (*i.e.,* LLaMA and BERT) but also significant in vision transformer models. Since outlier channels are often orders of magnitude bigger than the rest, they will dominate the quantization precision of the quantized tensor, resulting in less representation capacity for those channels with smaller magnitudes (Xiao et al., 2022). This makes tensor-wise or token-wise scaling factor insufficient for accurate activations quantization.

**Algorithm 1** FPQ baseline

1: **Input:** Calibration dataset, Full-precision Model $M$, Quantization format search space $R_X$ (e.g., $R_X = \{E3M0, E2M1, E1M2\}$ for FP4), number of round $n = 3$,
2: **Output:** FP $q$ Quantized model
3: **for** $l$ in $1^{st}$ to $L^{th}$ layer in $M$ **do**
4:     Forward & collect raw output $O^l = X^l Y^l$ of layer $l$;
5: **end for**
6: **for** $l$ in $1^{st}$ to $L^{th}$ layer in $M$ **do**
7:     Initialize the FP format search space w.r.t $X^l$ and $Y^l$ as $R_{\mathbf{X}} = \{r_{\mathbf{X}}^1, r_{\mathbf{X}}^2, ..., r_{\mathbf{X}}^t\}$ and $R_{\mathbf{Y}} = \{r_{\mathbf{Y}}^1, r_{\mathbf{Y}}^2, ....r_{\mathbf{Y}}^t\}$.
8:     Initialize bias $\tilde{b}_{\mathbf{X}}^i, \tilde{b}_{\mathbf{Y}}^i$ with Eq.7 for each format candidate $r_X^i \in R_{\mathbf{X}}$ and $r_{\mathbf{Y}}^i \in R_{\mathbf{Y}}$.
9:     Generate search space of $\tilde{b}_{\mathbf{X}}$ in $t$ formats to be $[\gamma_1 \tilde{b}_{\mathbf{X}}^{init}, \gamma_2 \tilde{b}_{\mathbf{X}}^{init}]$ and $\tilde{b}_{\mathbf{Y}}$ to be $[\gamma_1 \tilde{b}_{\mathbf{Y}}^{init}, \gamma_2 \tilde{b}_{\mathbf{Y}}^{init}]$.
10:     **for** 0 to n **do**
11:         Search for $\tilde{b}_{\mathbf{X}}^i$ w.r.t each $r_{\mathbf{X}}^i$ that minimizes Eq.12
12:         Search for $r_{\mathbf{X}}^i \in R_{\mathbf{X}}$ that minimizes Eq.12
13:         Search for $\tilde{b}_{\mathbf{Y}}^i$ w.r.t each $r_{\mathbf{Y}}^i$ that minimizes Eq.12
14:         Search for $r_{\mathbf{Y}}^i \in R_{\mathbf{Y}}$ that minimizes Eq.12
15:     **end for**
16: **end for**

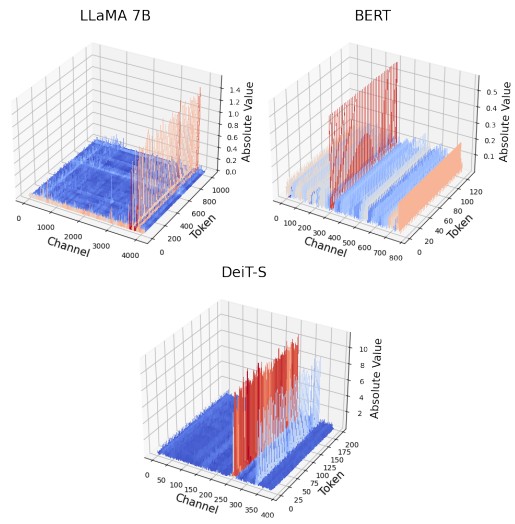

Figure 2: Magnitude of the output activations of the feed-forward network blocks in LLaMA-7B, BERT, and DeiT.

However, applying per-channel scaling factors for activations poses challenges to efficient matrix multiplication, because the scaling factor is not a shared constant along the multiplication direction and cannot be extracted as Eq. 10. To address this challenge, we introduce *pre-shifted exponent bias*, which allows us to calculate per-channel scaling factors from activations. These scaling factors are then re-parameterized as the exponent biases of the corresponding weights. This method effectively handles high inter-channel variance while maintaining nearly identical efficiency to per-tensor quantization.

Recalling in Eq. 7, we extracted the tensor-wise integer exponent bias $b$ and times it with real-valued scaling factor $\alpha$ and becomes a new scaling factor $\tilde{\alpha} = 2^{-\tilde{b}} = 2^{-b} \cdot \alpha$. Then, the floating-point quantization formula in Eq. 13 becomes:

$$X_{\mathrm{FP}} = 2^{-\tilde{b}}(-1)^s 2^{p-0}(1 + \frac{d_1}{2} + \frac{d_2}{2^2} + ... + \frac{d_m}{2^m}) \quad (13)$$

We note that after the bias is absorbed in the scaling factor, the original bias term ($b^{ori}$) in the FP formula is always zero. In dealing with the inter-channel variance, we devise an innovative usage of this integer exponent bias: we set it to be a per-channel variant ($\mathbf{b}^{ori} \in \mathbb{Z}^c$).

Then the calculation of the channel-wise integer bias vector ($\mathbf{b}^{ori}$) is very straightforward. We first calculate the initial per-channel real-valued scaling factor ($2^{-\tilde{\mathbf{b}}_j}$) from the per-channel maximum values:

$$\tilde{\mathbf{b}}_j = 2^e - \log_2(\max(|\mathbf{X}_{\mathrm{R}}^{:,j}|)) + \log_2(2 - 2^{-m}) - 1 \quad (14)$$

Here $\mathbf{X}_{\mathrm{R}}^{:,j}$ denotes the $j^{th}$ channel in the activation matrix. Then we separate $\tilde{\mathbf{b}}$ to a tensor-wise real-valued scaling factor plus a channel-wise integer scaling factor:

$$\begin{aligned} \tilde{\mathbf{b}} &= \tilde{\rho} + \mathbf{b}^{ori} \\ &= \tilde{\rho} + clip(\lfloor \tilde{\mathbf{b}} - \tilde{\rho} \rceil, 0, 2^{e-1}) \end{aligned} \quad (15)$$

where $\tilde{\rho} \in \mathbb{R}^1$, $\mathbf{b}^{ori} \in \mathbb{Z}^c$. Then the formula for one of the entries in the $j^{th}$ channel of $\mathbf{X}$ can be rewrote as follows:

$$\begin{aligned} X_{\mathrm{FP}} &= 2^{-\tilde{\mathbf{b}}_j}(-1)^s 2^{p-0}(1 + \frac{d_1}{2} + ... + \frac{d_m}{2^m}) \\ &= 2^{-\tilde{\rho}}(-1)^s 2^{p-\mathbf{b}_j^{ori}}(1 + \frac{d_1}{2} + ... + \frac{d_m}{2^m}) \end{aligned} \quad (16)$$

Note that the bias $\mathbf{b}^{ori}$ is constrained to integers within $[0, 2^e - 1]$, compatible with the standard floating-point number calculation. Nevertheless, adding different biases for each channel during inference may still cause some extra hardware operations. Thus, we re-parameterized the per-channel activation bias into a weight tensor and pre-computed the weights using the calibration set. This way, the exponent biases shifting only happens in the calibration stage. Then, an element in $j^{th}$ channel of activation tensors $X$ becomes:

$$X_{\mathrm{FP}} = 2^{-\tilde{\rho}}(-1)^s 2^{p-0}(1 + \frac{d_1}{2} + ... + \frac{d_m}{2^m}) \quad (17)$$

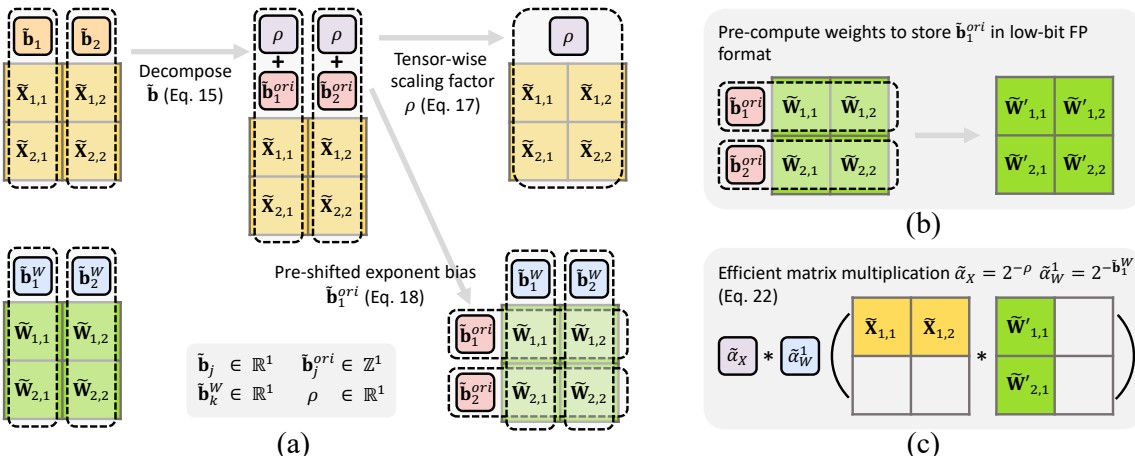

(a)

(b)

(c)

Figure 3: **Overview of pre-shifted exponent bias method**: **(a) Search phase**: The real-valued channel-wise scaling exponent bias for activations ($\tilde{\mathbf{b}}_j$) is partitioned into a real-valued tensor-wise exponent bias ($\rho$), and the integer-based channel-wise exponent bias ($\tilde{\mathbf{b}}_j^{ori}$). **(b) Reparameterization and weight pre-computation**: Once the optimal values are determined on the calibration set, $\tilde{\mathbf{b}}_j^{ori}$ are re-parameterized into the weight tensor. The weights are pre-computed to apply the bias, therefore this is a one-time cost. **(c) Inference phase**: The method leverages efficient matrix multiplication between low-bit floating-point matrices.

and the corresponding weight element in $j^{th}$ row of the weight tensor $W$ becomes:

$$W_{\text{FP}} = 2^{-\tilde{\mathbf{b}}^W}(-1)^s 2^{p-\mathbf{b}_j^{ori}}\left(1 + \frac{d_1}{2} + ... + \frac{d_m}{2^m}\right) \quad (18)$$

As result, efficient matrix multiplication in Eq.10 is reformulated as:

$$\mathbf{O}_{out}^{i,k} = \mathbf{X}_{\text{FP}}^{i,:}\mathbf{W}_{\text{FP}}^{:,k} = \tilde{\alpha}_{\mathbf{x}}\tilde{\alpha}_{\mathbf{w}}^k \tilde{\mathbf{X}}_{\text{FP}}^{i,:}(\beta \odot \tilde{\mathbf{W}}_{\text{FP}}^{:,k}) \quad (19)$$

where $\odot$ is the element-wise multiplication, $\beta = 2^{-\mathbf{b}^{ori}}$ and $(\beta \odot \tilde{\mathbf{W}}_{\text{FP}}^{:,k})$ can be pre-calculated and stored in low-bit FP format. We depict the overall *pre-shifted exponent bias* method in Fig.3. This method applies to quantizing all the fully-connected layers. During the search process, we initialize $\tilde{\rho}_{\mathbf{x}}$ as the $\min_j(\tilde{\mathbf{b}}_j)$. Then, we fixed $\tilde{\mathbf{b}}_{\mathbf{x}}$ to be the bias calculated from the Eq. 14 and search for the optimal $\tilde{\rho}_{\mathbf{x}}$ from $[\gamma_1 \tilde{\rho}_{\mathbf{x}}^{init}, \gamma_2 \tilde{\rho}_{\mathbf{x}}^{init}]$.

Combining *pre-shifted exponent bias* method with the joint format and max-value search framework(FPQ baseline), we name our method as (FPQ), short for *Floating Point Quantization*.

## 5 Experiments

To validate the effectiveness of the proposed method, we conduct experiments on LLaMA (Touvron et al., 2023) and BERT (Devlin et al., 2019) models in 5.2.1 and Sections 5.2.2. Further, in Section 5.2.3 we show that our method also generalizes well to vision transformer architectures. We present ablation studies on the calibration size

and search range in Section 5.3, and analyze the hardware costs of implementing FP operators in Section 5.4.

### 5.1 Experiments Details

We adopt per-tensor quantization for activation and per-channel quantization for weight. We employ layer reconstruction following the settings of (Yuan et al., 2022; Nagel et al., 2020), and parallel quantization based on the approach outlined in (Bai et al., 2022; Yuan et al., 2022). A more detailed discussion regarding our implementation decisions can be found in Appendix F. For LLaMA models, we quantize all the weight and activation tensors in fully-connected layers for a fair comparison with previous work (Xiao et al., 2022; Liu et al., 2023). For BERT and ViT models, both fully-connected layers and activation-activation multiplication tensors in the self-attention module are quantized. Note that for FPQ on BERT (Devlin et al., 2019) and ViTs models, the reconstruction metric Eq. 12 is substituted with a Hessian approximation loss metric. This substitution is further detailed in Appendix A.

### 5.2 Main Results

#### 5.2.1 LLM Zero-Shot Reasoning

We evaluate the effectiveness of FPQ for LLaMA-7B/ LLaMA-13B (Touvron et al., 2023) on common sense zero-shot reasoning tasks. For the calibration data, we sample 32 random segments with 2048 tokens length from the C4 (Raffel et al., 2020)

| Quant Method | #Bits (E/W/A) | # Calib | BoolQ | PIQA | HellaSwag | WinoGrande | ARC-e | ARC-c | Avg. |
|---|---|---|---|---|---|---|---|---|---|
| LLaMA-7B Full-precision | 16/16/16 | - | 75.1 | 78.7 | 56.9 | 69.9 | 75.3 | 41.9 | 66.3 |
| MinMax INT Quant | 8/8/8 | 32 | 64.3 | 66.8 | 40.5 | 57.4 | 59.0 | 29.6 | 52.9 |
| MinMax FP Quant (E4M3) | 8/8/8 | 32 | 74.9 | 78.6 | 56.8 | 69.5 | 75.5 | 41.6 | **66.1** |
| SmoothQuant (Xiao et al., 2022) | 16/8/8 | 512 | 74.0 | 77.5 | 55.0 | 69.6 | 74.4 | 37.4 | 64.6 |
| FPQ baseline | 8/8/8 | 32 | 75.8 | 78.3 | 55.9 | 69.5 | 75.6 | 41.3 | **66.1** |
| FPQ | 8/8/8 | 32 | 75.6 | 78.2 | 56.6 | 70.2 | 74.6 | 40.7 | 66.0 |
| MinMax INT Quant | 4/4/16 | 32 | 64.1 | 76.1 | 51.6 | 66.3 | 72.4 | 40.0 | 61.7 |
| MinMax FP Quant (E2M1) | 4/4/16 | 32 | 73.0 | 77.9 | 55.2 | 69.1 | 73.6 | 40.9 | 64.9 |
| GPTQ (Frantar et al., 2023) | 4/4/16 | 128 | 73.3 | 77.9 | 54.9 | 67.9 | 72.7 | 37.4 | 64.0 |
| FPQ baseline | 4/4/16 | 32 | 74.8 | 77.9 | 55.6 | 69.5 | 75.2 | 41.0 | **65.7** |
| FPQ | 4/4/16 | 32 | 74.2 | 77.8 | 55.8 | 69.9 | 74.9 | 40.4 | 65.5 |
| MinMax INT Quant | 4/4/8 | 32 | 50.4 | 56.5 | 27.9 | 46.5 | 36.1 | 21.2 | 39.7 |
| MinMax FP Quant (E2M1/E4M3) | 4/4/8 | 32 | 73.0 | 77.5 | 55.0 | 69.3 | 73.6 | 40.9 | 64.9 |
| FPQ baseline | 4/4/8 | 32 | 75.0 | 77.6 | 55.9 | 69.9 | 74.3 | 39.4 | 65.3 |
| FPQ | 4/4/8 | 32 | 75.0 | 77.7 | 55.5 | 69.8 | 74.5 | 39.9 | **65.4** |
| MinMax INT Quant | 4/4/4 | 32 | 54.1 | 51.7 | 25.6 | 49.8 | 24.7 | 22.9 | 38.1 |
| MinMax FP Quant (E2M1) | 4/4/4 | 32 | 47.3 | 53.1 | 25.7 | 50.7 | 25.1 | 22.4 | 37.4 |
| SmoothQuant (Xiao et al., 2022) | 16/4/4 | 512 | 54.1 | 62.8 | 41.5 | 52.6 | 50.6 | 32.9 | 49.1 |
| LLM-QAT (Liu et al., 2023) | 16/4/4 | (QAT) | 63.5 | 64.3 | 55.6 | 52.9 | 50.3 | 30.2 | 52.8 |
| FPQ baseline | 4/4/4 | 32 | 57.4 | 56.6 | 30.2 | 51.1 | 37.7 | 23.2 | 42.7 |
| FPQ | 4/4/4 | 32 | 64.2 | 73.5 | 47.8 | 63.7 | 65.9 | 33.6 | **58.1** |
| LLaMA-13B Full-precision | 16/16/16 | - | 77.9 | 79.2 | 59.9 | 72.6 | 77.4 | 46.4 | 68.9 |
| MinMax INT Quant | 8/8/8 | 32 | 60.6 | 69.6 | 46.0 | 61.5 | 63.3 | 32.8 | 55.6 |
| MinMax FP Quant (E4M3) | 8/8/8 | 32 | 78.0 | 79.1 | 60.0 | 72.3 | 77.2 | 47.1 | **68.9** |
| SmoothQuant (Xiao et al., 2022) | 16/8/8 | 512 | 76.5 | 78.0 | 58.0 | 72.1 | 76.3 | 45.5 | 68.2 |
| FPQ baseline | 8/8/8 | 32 | 78.0 | 79.1 | 59.9 | 72.3 | 77.2 | 47.1 | **68.9** |
| FPQ | 8/8/8 | 32 | 78.1 | 78.5 | 59.1 | 72.4 | 76.4 | 46.1 | 68.4 |
| MinMax INT Quant | 4/4/8 | 32 | 52.1 | 65.0 | 36.4 | 53.9 | 52.3 | 29.0 | 48.1 |
| MinMax FP Quant (E2M1/E4M3) | 4/4/8 | 32 | 78.0 | 78.9 | 58.0 | 71.6 | 76.0 | 44.8 | **67.9** |
| FPQ baseline | 4/4/8 | 32 | 76.2 | 78.2 | 57.9 | 71.9 | 75.1 | 43.9 | 67.2 |
| FPQ | 4/4/8 | 32 | 76.4 | 78.5 | 58.2 | 72.1 | 75.2 | 44.7 | 67.5 |
| MinMax INT Quant | 4/4/4 | 32 | 54.5 | 52.7 | 25.5 | 51.1 | 25.3 | 22.1 | 38.5 |
| MinMax FP Quant (E2M1) | 4/4/4 | 32 | 45.8 | 51.7 | 25.5 | 49.5 | 25.0 | 22.8 | 36.7 |
| SmoothQuant (Xiao et al., 2022) | 16/4/4 | 512 | 57.6 | 61.3 | 56.0 | 52.6 | 49.9 | 25.1 | 50.4 |
| FPQ baseline | 4/4/4 | 32 | 54.3 | 57.7 | 35.7 | 52.2 | 41.1 | 25.7 | 44.5 |
| FPQ | 4/4/4 | 32 | 71.9 | 74.8 | 53.3 | 66.7 | 71.7 | 39.9 | **63.1** |

Table 1: Zero-shot performance on common sense reasoning tasks with LLaMA (Touvron et al., 2023) models. We denote E/W/A as the bit-width of word embeddings, model weight and activations, respectively.

dataset following the setting of GPTQ (Frantar et al., 2023). The data preprocessing and score calculation are based on EleutherAI evaluation harness[1]. In Table 1, we compare FPQ to the floating-point PTQ baselines, and state-of-the-art PTQ and QAT methods, including SmoothQuant (Xiao et al., 2022) and GPTQ (Frantar et al., 2023), and LLM-QAT (Liu et al., 2023).

In general, all methods, except for the naïve Min-Max INT Quantization, produce comparable outcomes in the 8-bit setting on both LLaMA-7B and LLaMA-13B. Additionally, we observe that the naïve MinMax FP Quantization achieves nearly lossless results and even surpasses the state-of-the-art integer post-training quantization method, SmoothQuant (Xiao et al., 2022), which indicates that floating-point quantization naturally has a strong capability in handling the distributions in transformers. However, both MinMax FP Quant and FPQ baseline fail when pushing the quan-

tization precision to ultra-low 4/4/4 bit setting, with 28.9% and 23.8% accuracy degradation on LLaMA-7B, respectively. In this extreme case, the previous state-of-the-art PTQ and QAT methods, SmoothQuant (Xiao et al., 2022) and LLM-QAT (Liu et al., 2023) also suffer severe accuracy downgrade. In comparison, FPQ demonstrates a strong capability of handling extra-low bit settings and achieves only 8.2/5.8% accuracy drop on LLaMA-7B/13B with 4/4/4 bit-width, outperforming SmoothQuant (Xiao et al., 2022) by a large margin, yet with less bit-width and smaller calibration size. Moreover, FPQ even achieves 5.3% accuracy improvements compared to LLM-QAT (Liu et al., 2023) in the 4/4/4 setting and 1.5% over GPTQ (Frantar et al., 2023) in the 4/4/16 configuration on LLaMA-7B.

For practitioners, a crucial consideration is determining the appropriate quantization methods for various bit-widths. Therefore, based on our findings, we offer two recommendations that balance the trade-off between accuracy and

[1]https://github.com/EleutherAI/lm-evaluation-harness

| Quant Method | #Bits (E/W/A) | # Calib | MNLI$_{-m}$ | QQP | QNLI | SST-2 | CoLA | STS-B | MRPC | RTE | Avg. |
|---|---|---|---|---|---|---|---|---|---|---|---|
| (Full-precision) | 32-32-32 | - | 84.9 | 91.4 | 92.1 | 93.2 | 59.7 | 90.1 | 86.3 | 72.2 | 83.7 |
| MinMax INT Quant | 8/8/8 | 128 | 77.0 | 89.9 | 88.9 | 92.9 | 51.8 | 88.2 | 83.8 | 71.5 | 80.5 |
| MinMax FP Quant (E2M5) | 8/8/8 | 128 | 78.9 | 90.8 | 88.6 | 92.9 | 52.7 | 88.4 | 84.3 | 69.0 | 80.7 |
| MinMax FP Quant (E3M4) | 8/8/8 | 128 | 84.5 | 90.9 | 91.5 | 93.2 | 58.3 | **89.3** | 87.7 | 71.8 | 83.4 |
| MinMax FP Quant (E4M3) | 8/8/8 | 128 | **84.7** | 90.9 | **91.7** | 93.0 | 58.6 | **89.3** | 86.5 | **72.2** | 83.4 |
| MinMax FP Quant (E5M2) | 8/8/8 | 128 | 84.1 | 90.9 | 91.4 | **93.6** | 58.1 | 89.2 | 87.5 | 71.8 | 83.3 |
| FPQ baseline | 8/8/8 | 128 | 84.6 | 90.9 | **91.7** | 93.1 | 58.6 | 89.3 | **88.0** | **72.2** | 83.5 |
| FPQ | 8/8/8 | 128 | 84.6 | **91.0** | 91.6 | 93.3 | **58.8** | 89.3 | **88.0** | **72.2** | **83.6** |
| MinMax INT Quant | 6/6/6 | 128 | 31.9 | 62.0 | 52.8 | 58.8 | 0.0 | 12.7 | 32.1 | 52.7 | 37.9 |
| MinMax FP Quant (E2M3) | 6/6/6 | 128 | 43.5 | 85.4 | 79.4 | 90.5 | 45.2 | 86.0 | 66.9 | 59.9 | 69.6 |
| MinMax FP Quant (E3M2) | 6/6/6 | 128 | 83.9 | 90.8 | 90.8 | 92.2 | 58.2 | 88.6 | 87.0 | **72.2** | 83.0 |
| MinMax FP Quant (E4M1) | 6/6/6 | 128 | 84.4 | 90.2 | 90.1 | 92.2 | 58.2 | 89.2 | 85.3 | 69.7 | 82.4 |
| FPQ baseline | 6/6/6 | 128 | **84.6** | **90.9** | 91.2 | **93.2** | **58.8** | 88.7 | 87.5 | 70.8 | **83.2** |
| FPQ | 6/6/6 | 128 | 84.5 | 90.8 | **91.6** | 93.1 | 57.3 | **89.3** | **88.7** | 71.8 | **83.2** |
| MinMax INT Quant | 4/4/8 | 128 | 33.1 | 63.8 | 60.1 | 49.3 | 0.0 | 44.0 | 50.2 | 49.1 | 43.7 |
| MinMax FP Quant (E2M1) | 4/4/8 | 128 | 60.6 | 70.9 | 77.4 | 79.9 | 5.5 | 78.6 | 46.8 | 56.6 | 59.5 |
| MREM-S (Bai et al., 2022) | 4/4/8 | 4096 | 83.5 | 90.2 | **91.2** | 91.4 | 55.1 | 89.1 | 84.8 | 71.8 | 82.1 |
| MREM-P (Bai et al., 2022) | 4/4/8 | 4096 | 83.4 | 90.2 | 91.0 | 91.5 | 54.7 | 89.1 | 86.3 | 71.1 | 82.2 |
| FPQ baseline | 4/4/8 | 128 | 84.4 | **90.6** | **91.4** | **92.9** | 58.6 | 83.7 | 88.2 | **73.3** | 82.9 |
| FPQ | 4/4/8 | 128 | **84.5** | **90.6** | 91.1 | 92.7 | **58.8** | **89.3** | **88.7** | **73.3** | **83.6** |
| MinMax INT Quant | 4/4/4 | 128 | 31.8 | 39.7 | 50.5 | 49.1 | 0.0 | 6.7 | 31.6 | 54.5 | 32.9 |
| MinMax FP Quant (E2M1) | 4/4/4 | 128 | 33.6 | 54.0 | 50.6 | 50.8 | 0.0 | 0.0 | 31.6 | 52.0 | 34.1 |
| BrecQ (Li et al., 2021) | 8/4/4 | 4096 | 31.9 | 62.3 | 50.7 | 50.9 | 0.9 | 6.4 | 31.7 | 52.3 | 35.8 |
| QDrop (Wei et al., 2022) | 8/4/4 | 4096 | 71.4 | 79.0 | 76.8 | 88.1 | 40.9 | 81.9 | 79.2 | 60.7 | 72.3 |
| FPQ baseline | 4/4/4 | 128 | 38.9 | 68.3 | 55.3 | 83.6 | 10.6 | 0.0 | 43.8 | 55.2 | 44.5 |
| FPQ | 4/4/4 | 128 | **82.3** | **89.2** | **86.6** | **91.5** | **52.6** | **85.5** | **83.8** | **69.0** | **80.1** |

Table 2: Results on the GLUE development set with BERT (Bai et al., 2022) model. We denote E/W/A as the bit-width of word embeddings, model weight and activations, respectively.

search/optimization efficiency. First of all, since the difference between MinMax FP Quant and the rest of the methods is marginal for the 8/8/8 setting, we recommend simply using the MinMax FP Quant method for the 8/8/8 setting as the MinMax method does not involve search process. However, for more demanding scenarios, especially with activation quantization to 4 bits, we recommend employing FPQ for minimizing accuracy degradation with negligible inference overhead.

### 5.2.2 BERT Model

We evaluate the proposed quantization techniques for BERT model on GLUE tasks (Wang et al., 2019). Full-precision BERT-base models finetuned on GLUE datasets are obtained from Huggingface public repository[2]. We randomly sample 128 data from the training set as the calibration set. In Table 2, FPQ demonstrates remarkable performance, achieving absolute average accuracy improvements of $44.3\%$ compared to BrecQ (Li et al., 2021) and $7.9\%$ over QDrop (Wei et al., 2022) with 4/4/4 bit setting. Further, with 4-bit weight and 8-bit activation, MREM-S/MREM-P (Bai et al., 2022) present a 1.6/1.5% accuracy gap to the full-precision model with 4096 calibration data, while FPQ achieves almost no accuracy loss with only

[2]https://huggingface.co/textattack/bert-base-uncased-{TASK_NAME}

128 calibration data points.

### 5.2.3 Generalizability on Vision Transformer

Based on our findings that vision transformers also exhibit a consistent activation distribution pattern as language transformers, characterized by high inter-channel variance and low intra-channel variance, as detailed in Fig. 2, we extended our proposed methods to ViT and compared FPQ with floating-point PTQ baselines and state-of-the-art PTQ method for ViT on the ImageNet classification task. Table 3 shows that findings on ViT are consistent with that on language models: previous state-of-the-art integer-based methods struggled to maintain reasonable accuracy when quantizing the transformer to lower bits. In comparison, the proposed FPQ outperformed both PTQ4ViT and APQ-ViT on 6 bits, and also achieved 40.9% and 31.5% absolute accuracy improvement over PTQ4ViT and APQ-ViT on DeiT-S in the 4-bit configuration.

### 5.3 Ablation Study

In this section, we first compare the influence of different calibration sizes on FPQ. We vary the calibration size in $\{32, 64, 128, 256\}$ and test on MNLI, QQP, and CoLA. Table 4 shows that the evaluation on MNLI and QQP is more robust to different settings, and the variance is more significant on CoLA. We observe that FPQ performs well with a

| W/A | Quant Method | Deit-S | Deit-B | ViT-S |
|---|---|---|---|---|
| Full-prec | - | 79.9 | 81.8 | 81.4 |
| 6/6 | PTQ4ViT(Yuan et al., 2022) | 76.3 | 80.3 | 78.6 |
| 6/6 | APQ-ViT(Ding et al., 2022) | 77.8 | 80.4 | 79.2 |
| 6/6 | MinMax FP Quant (E3M2) | 79.3 | 81.7 | 80.7 |
| 6/6 | FPQ baseline | 79.43 | 81.7 | 80.9 |
| 6/6 | FPQ | **79.5** | **81.8** | **81.1** |
| 4/4 | PTQ4ViT(Yuan et al., 2022) | 34.1 | 64.4 | 42.6 |
| 4/4 | APQ-ViT (Ding et al., 2022) | 43.6 | 67.5 | 48.0 |
| 4/4 | MinMax FP Quant (E2M1) | 0.4 | 0.1 | 0.1 |
| 4/4 | FPQ baseline | 6.57 | 0.71 | 0.3 |
| 4/4 | FPQ | **75.0** | **79.4** | **73.2** |

Table 3: Comparison on the ImageNet dataset with vision transformer structures.

| E/W/A | #Calib | MNLI-M | QQP | CoLA |
|---|---|---|---|---|
| 4/4/4 | 32 | 81.5 | **89.4** | 44.4 |
| 4/4/4 | 64 | 81.8 | 89.4 | 47.9 |
| 4/4/4 | 128 | **82.3** | 89.2 | 52.6 |
| 4/4/4 | 256 | 81.9 | 89.0 | **52.9** |
| 6/6/6 | 32 | **84.8** | 90.8 | 55.0 |
| 6/6/6 | 64 | 84.7 | **90.9** | **58.2** |
| 6/6/6 | 128 | 84.5 | 90.8 | 57.3 |
| 6/6/6 | 256 | 84.6 | 90.8 | 57.6 |

Table 4: Ablation studies of different calibration sizes.

calibration set size of 128 data points. However, we also find that it remains robust and maintains competitive accuracy even with limited access to calibration data, such as when using as few as 32 data points.

We investigate the robustness of FPQ to different search ranges $(\gamma_1, \gamma_2)$. Table 5 presents the results of FPQ using three sets of $(\gamma_1, \gamma_2)$: $(0.01, 1.2), (0.1, 1.2), (0.5, 1.5)$, on MNLI, QQP, and CoLA. It is observed that no single search range outperforms the others consistently across all tasks. For instance, the search range $(0.01, 1.2)$ performs better than $(0.5, 1.5)$ on MNLI and QQP, but slightly worse on CoLA in the 4-bit configuration. Overall, FPQ exhibits robustness to various $\gamma_1$ and $\gamma_2$, as long as the search range is not overly aggressive.

## 5.4 Hardware Cost

We further examine the hardware utilization of low-bit INT, FP, and mixed-format FP multiplication operators, including adder, multiplier, and multiply-accumulate (MAC) units, in terms of hardware area. Mixed-format FP refers to the multiplication of floating-point numbers with different formats, *e.g.*, E2M1 multiplies with E1M2. We implemented the MAC operator by Verilog HDL and utilized Cadence Genus to obtain the synthesized area under TSMC 40nm technology and 0.5GHz clock frequency.

Table 6 illustrates the hardware cost of the INT and FP operators, with the multiplier being the pri-

| E/W/A | $\gamma_1, \gamma_2$ | MNLI-M | QQP | CoLA |
|---|---|---|---|---|
| 4/4/4 | 0.01, 1.2 | **82.3** | **89.2** | 52.6 |
| 4/4/4 | 0.1, 1.2 | 82.2 | 89.1 | **53.6** |
| 4/4/4 | 0.5, 1.5 | 82.3 | 88.4 | 52.8 |
| 6/6/6 | 0.01, 1.2 | 84.5 | 90.8 | 57.3 |
| 6/6/6 | 0.1,1.2 | **84.7** | 90.8 | 57.5 |
| 6/6/6 | 0.5,1.5 | 84.7 | **90.8** | **57.8** |

Table 5: Ablation studies of different search range.

| Format | Adder($\mu m^2$) | Multiplier($\mu m^2$) | MAC($\mu m^2$) |
|---|---|---|---|
| INT4 | 93 | 182 | 410 |
| INT6 | 132 | 340 | 529 |
| E2M1 | 111 | 92 | 443 |
| E3M2 | 223 | 138 | 498 |
| E2M1 * E1M2 | 105 | 107 | 432 |

Table 6: Area differences of INT, FP and mixed Format FP operators across different bit-widths.

mary cost for INT and the adder for FP. Notably, the disparity between FP4 and INT4 adders is small, while INT has twice the hardware cost for the multiplier. Moreover, the mixed-format FP4 operator has comparable hardware area as the standard FP4 operator. These findings indicate that the proposed FPQ approach imposes negligible overhead in terms of hardware implementation when compared to the standard FP operators and the hardware cost for FP is comparable with INT.

## 6 Conclusion

This paper presents the first successful demonstration of 4-bit floating-point post-training quantization for weights, activations, and embeddings in natural language transformer architectures, including both large language models and BERT model. We also extend our method to vision transformers and observe its robust generalization ability. Our approach involves a practical search-based technique which establishes a strong baseline and achieves state-of-the-art results for 6-bit and 8-bit quantization. Furthermore, we address the challenge of high inter-channel variance in transformers by proposing *pre-shifted exponent bias*, which proves highly effective in achieving accurate 4-bit quantization.

## Acknowledgement

This research is supported by National Natural Science Foundation of China/ HKSAR Research Grants Council Joint Research Scheme under Grant $NHKUST627/20$, and Foshan HKUST Projects under Grant $FSUST21 - HKUST10E$.

## Limitations

Our experiments were conducted on publicly available datasets with finite sentence lengths, and the generalizability of our method to extremely long sequences or streaming data has not been verified and may require further investigation. In addition, it remains to be seen how our proposed method can generalize to other domains beyond language and vision, such as audio. It would also be interesting to see the applicability of our method to generative tasks and other applications.

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

## A Hessian-Based Loss Metric

The objective of post-training quantization is to minimize the perturbation ($\delta\mathbf{X} = \mathbf{X}_{\text{FP}} - \mathbf{X}_{\text{R}}$) introduced by quantization to the pre-trained real-valued network:

$$\min \mathbb{E}[\mathcal{L}(\mathbf{X}_{\text{R}} + \delta\mathbf{X}) - \mathcal{L}(\mathbf{X}_{\text{R}})] \qquad (20)$$

Following the Taylor series expansion, we have

$$\mathbb{E}[\mathcal{L}(\mathbf{X}_{\text{R}} + \delta\mathbf{X}) - \mathcal{L}(\mathbf{X}_{\text{R}})]$$
$$\approx \delta\mathbf{X}^T \bar{\mathbf{g}}^{(\mathbf{X})} + \frac{1}{2}\delta\mathbf{X}^T \bar{\mathbf{H}}^{(\mathbf{X})}\delta\mathbf{X} \qquad (21)$$
$$\approx \frac{1}{2}\delta\mathbf{X}^T \bar{\mathbf{H}}^{(\mathbf{X})}\delta\mathbf{X}$$

Here, $\bar{\mathbf{g}}^{(\mathbf{X})}$ is the gradients and $\bar{\mathbf{H}}^{(\mathbf{X})}$ is the Hessian matrix. Since the pre-trained model is well-converged, we can assume that $\bar{\mathbf{g}}^{(\mathbf{X})}$ has near zero value in every element, and thus term $\delta\mathbf{X}^T\bar{\mathbf{g}}^{(\mathbf{X})}$ can be neglected.

The Hessian matrix $\bar{\mathbf{H}}^{(\mathbf{X})}$ is computed as:

$$\bar{\mathbf{H}}^{(\mathbf{X})} = \mathbf{J}_{\mathbf{O}}^T(\mathbf{X})\bar{\mathbf{H}}^{(\mathbf{O})}\mathbf{J}_{\mathbf{O}}(\mathbf{X}) \qquad (22)$$

where $\mathbf{J}_{\mathbf{O}}(\mathbf{X})$ denotes the Jacobian matrix of the layer output $\mathbf{O}$ *w.r.t* $\mathbf{X}$, and $\bar{\mathbf{H}}^{(\mathbf{O})}$ is the Hessian matrix *w.r.t* $\mathbf{O}$. We then substitute the above equation back to equation 21 :

$$\delta\mathbf{X}^T\bar{\mathbf{H}}^{(\mathbf{X})}\delta\mathbf{X}$$
$$= (\mathbf{J}_{\mathbf{O}}(\mathbf{X})\delta\mathbf{X})^T\bar{\mathbf{H}}^{(\mathbf{O})}(\mathbf{J}_{\mathbf{O}}(\mathbf{X})\delta\mathbf{X}) \qquad (23)$$
$$\approx (\hat{\mathbf{O}} - \mathbf{O})^T\bar{\mathbf{H}}^{(\mathbf{O})}(\hat{\mathbf{O}} - \mathbf{O})$$

Here $\hat{\mathbf{O}}$ is the intermediate output of the quantized layer and $\mathbf{O}$ is the original layer output. Note that under the assumption that $\delta\mathbf{X}$ is relatively small (Li et al., 2021), we can approximate $(\hat{\mathbf{O}} - \mathbf{O})$ as $\mathbf{J}_{\mathbf{O}}(\mathbf{X})\delta\mathbf{X}$ using first-order Taylor expansion.

Nevertheless, the calculation of $\bar{\mathbf{H}}^{(\mathbf{O})}$ is still burdensome, therefore, we use the diagonal entries of the Fisher Information Matrix of $\mathbf{O}$ to substitute $\bar{\mathbf{H}}^{(\mathbf{O})}$ following (Li et al., 2021; Yuan et al., 2022), and the new Hessian-based metric becomes:

$$\mathbb{E}[(\hat{\mathbf{O}} - \mathbf{O})^T diag((\frac{\partial L}{\partial \mathbf{O}_1})^2, ..., (\frac{\partial L}{\partial \mathbf{O}_n})^2(\hat{\mathbf{O}} - \mathbf{O})]$$
$$(24)$$

Here, each entry of $\mathbf{O}$ is assumed to be independent and $n$ denoted the total number of elements in $\mathbf{O}$. In this study, this hessian-based metric is used as the reconstruction metric to search for the optimal FP quantization function for both the weight and activation when performing layer-wise reconstruction in BERT and Vision Transformer models.

## B Quantization Error of Different Floating-Point Formats

Figure 4 compares the quantization error of different formats in 8-bit quantization, including INT8, E2M5, E3M4, E4M3, and E5M2. We apply these formats to different BERT modules in the first, fifth, and last layers. The figures demonstrate that the optimal FP formats differs depending on the specific module that we are quantizing.

## C Inter-Channel Variance Visualization

Figure 5 and 6 depict the output of different fully-connected layers in BERT for the MNLI task, DeiT-S for the ImageNet-1K task, and LLaMA-7B for the zero-shot reasoning task. The visualizations reveal a noticeable inter-channel variance presented in both language and vision transformers.

## D Efficient Matrix Multiplication

Figure 7 displays a comprehensive list of all the granularity options that allow for efficient matrix multiplication. While per-token quantization theoretically provides greater precision in terms of quantization granularity, the accuracy gains achieved through this method are minimal and do not justify the additional computational overhead required. As a result, we have opted to use per-tensor quantization when quantizing activations.

## E Learning Format and Maximum Value

We compare the previous gradient-based method (Kuzmin et al., 2022) with the proposed search-based method for finding the optimal format and maximum value. On DeiT-S, the learnable method only achieves 74.38% accuracy for an 8-bit quantized model on ImageNet, in contrast, FPQ can attain an almost loss-less result of 79.88%. We analyze the gradients for the number of exponent bits $e$ derived in (Kuzmin et al., 2022) and observe that each time the exponent bits change, the gradients experience exponential variations, leading to high instability. Based on this observation, we assert that employing a search-based method to determine the optimal formats is crucial in post-training quantization (PTQ).

## F Reconstruction Choices

The previous works on integer post-training quantization involves breaking down the target model into

sub-modules and reconstructing them separately (Nagel et al., 2020; Li et al., 2021; Bai et al., 2022; Yuan et al., 2022). This addresses the problem of over-fitting, given that only a limited amount of unlabeled calibration data is available. In this study we find the layer-wise reconstruction and parallel quantization works best for floating-point PTQ:

**Layer Reconstruction:** Recent research (Li et al., 2021; Bai et al., 2022) suggests increasing the reconstruction granularity from layer reconstruction (Nagel et al., 2020) to block reconstruction (Li et al., 2021) or even larger granularity (Lee et al., 2023). This is achieved by jointly optimizing all the linear layers or matrix multiplication components within each module to prevent the propagation of reconstruction errors among the layers. Despite this, we have observed that increasing the recon-struction granularity does not improve the accuracy of FPQ baseline or sometimes even lead to worse results. Therefore, we choose layer reconstruction.

**Parallel Quantization:** Sequential quantization is the most commonly used approach (Wu et al., 2020; Nagel et al., 2020; Li et al., 2021) where modules are quantized consecutively based on their sequential order, and the input for the current calibrating module is generated using all the previously quantized modules. However, some recent works (Yuan et al., 2022; Bai et al., 2022) proposed a new parallel quantization framework. This framework uses the raw output of the full-precision modules as input and makes the calibration of each module independent from one another. In this work, we use parallel quantization, as it yields better results than its sequential counterparts.

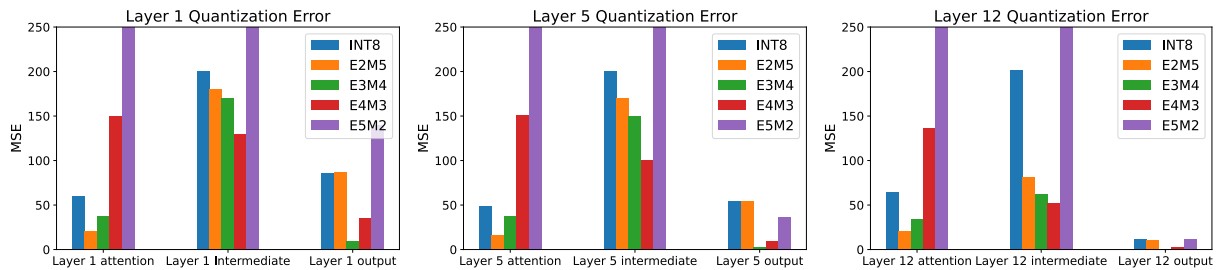

Figure 4: Quantization error of different formats for BERT layers.

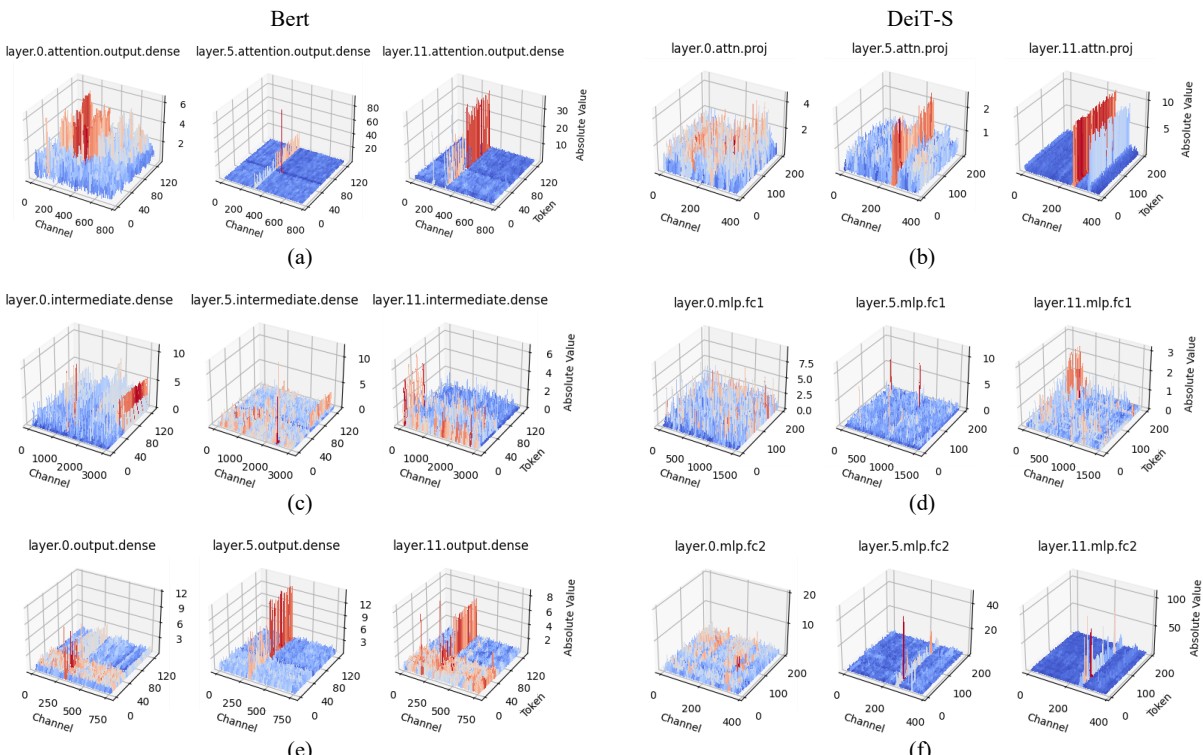

Figure 5: Magnitude of the output activations of different modules in BERT (left column), and DeiT-S (right column).

LLaMa-7B

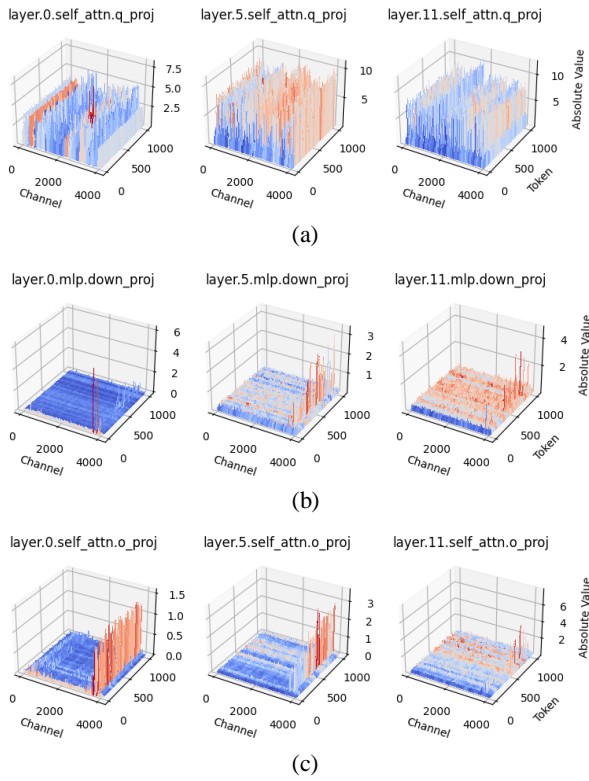

Figure 6: Magnitude of the output activations of different modules in LLaMA-7B.

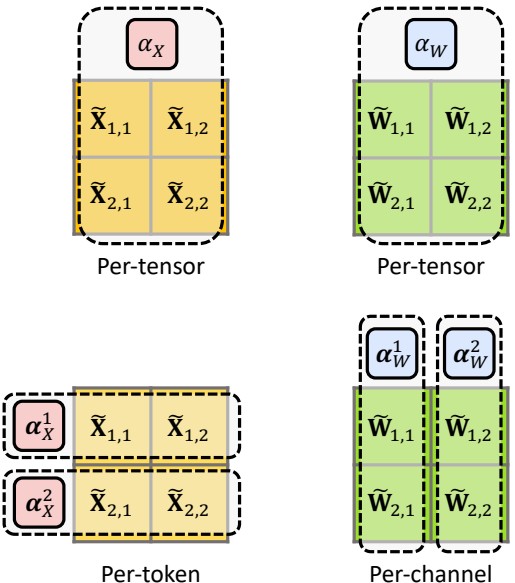

Figure 7: Quantization granularity options that support efficient matrix multiplication. The dimensions that share the same scaling factor are indicated with red dotted frames