# OpenReview forum: "LLM-FP4: 4-Bit Floating-Point Quantized Transformers"
_EMNLP/2023/Conference — EMNLP 2023 Main_

### Official Review · Reviewer_EuFY · 2023-07-25

**Soundness:** 3

**Excitement:**

4: Strong: This paper deepens the understanding of some phenomenon or lowers the barriers to an existing research direction.

**Paper Topic And Main Contributions:**

This paper presents a novel floating-point (FP) post-training quantization (PTQ) technique for transformer-based models. The authors integrate hessian-based loss into quantization parameter search to develop a strong FP-PTQ baseline. They further propose per-channel activation quantization and demonstrate that reparameterizing additional scaling factors as exponential biases of weights incurs minimal computational cost. The method achieves impressive compression, quantizing both weights and activations in BERT to 4 bits, with just a 3.6-point drop in average GLUE score compared to the full-precision model.

**Reasons To Accept:**

The paper is excellently written and highly accessible, even for readers outside the domain, thanks to the well-presented background on linear and floating-point quantization. The addressed problem carries significant importance within the NLP community. The proposed method is technically sound and demonstrates promising empirical performance across both language and vision benchmarks. Additionally, the authors' provision of ample implementation details facilitates easy reproduction by other researchers and practitioners in the community.

**Reasons To Reject:**

The paper would significantly benefit from an improved presentation of results. It is essential to specify whether the baselines in both Table 1 and Table 2 are using linear or floating-point quantization techniques. While the authors have included the results of the FPT baseline for 8/8/8 and 6/6/6 settings in Table 1, they should also include the results for the other settings in Table 1 and in the other tables. These additional results are pivotal in allowing readers to gain a comprehensive understanding of the effectiveness of the proposed pre-shifted exponent bias and its performance under various configurations.

While the paper is well-written overall, some readers may find it a bit dense in terms of the mathematical formulations. To address this concern, the inclusion of more accompanying figures and visual aids could significantly improve the paper's accessibility. Illustrating the key concepts and processes graphically would help bridge the gap between complex mathematical expressions and readers' comprehension, resulting in a more inclusive and engaging reading experience.

**Reproducibility:**

4: Could mostly reproduce the results, but there may be some variation because of sample variance or minor variations in their interpretation of the protocol or method.

**Reviewer Confidence:**

3: Pretty sure, but there's a chance I missed something. Although I have a good feel for this area in general, I did not carefully check the paper's details, e.g., the math, experimental design, or novelty.

---

> ### Author Rebuttal · Authors · 2023-08-28
>
> **[Weakness 1: More experiments on different configurations]**
>
> Based on your suggestion, we conducted additional experiments on both Bert and LLaMa-7B using the FPT baseline for the 4/4/4 setting. The results indicate that in 4-bit quantization, the FPT baseline is unable to maintain accuracy for such a low-bit setting due to significant variance across channels. In comparison, after utilizing the proposed pre-shifted exponent bias, FPT is capable of dealing with channel-wise variance and significantly improves accuracy. For 4/4/16 settings in Table 2, FPT is the same as FPT baseline since the activations are not quantized. We will add these results to the manuscript.
>
> |Methods | #Bits (E/W/A) | #Calib | $\textbf{MNLI}_{\rm -m}$ | $\textbf{QQP}$ |  $\textbf{QNLI}$ | $\textbf{SST-2}$ | $\textbf{CoLA}$ | $\textbf{STS-B}$ | $\textbf{MRPC}$ | $\textbf{RTE}$ | $\textbf{Avg.}$ |
> | -------- | -------- | -------- | -------- | -------- | -------- | -------- | -------- | -------- | -------- | -------- | -------- |
> | Bert (Full-precision) | 32-32-32 | - | 84.9 | 91.4 | 92.1 | 93.2 | 59.7 | 90.1 | 86.3 | 72.2 | 83.7 |
> |MinMax(E2M1)| 4/4/4 | 128 | 33.6 | 54.0 | 50.6 | 50.8 | 0.0 | 0.0 | 31.6 | 52.0 | 34.1 |
> | BrecQ | 8/4/4 | 4096 | 31.9 | 62.3 | 50.7 | 50.9 | 0.9 | 6.4 | 31.7 | 52.3 | 35.8 |
> | QDrop | 8/4/4 | 4096 | 71.4 | 79.0 | 76.8 | 88.1 | 40.9 | 81.9 | 79.2 | 60.7 | 72.3 |
> | FPT Baseline | 4/4/4 | 128 | 38.9 | 68.3 | 55.3 | 83.6 | 10.6 | 0.0 | 43.8 | 55.2 | 44.5 |
> | FPT | 4/4/4 | 128 | 82.3 | 89.2 | 86.6 | 91.5 | 52.6 | 85.5 | 83.8 | 69.0 | $\textbf{80.1}$ |
>
> | Quant Methods          | #Bits (E/W/A) |  #Calib | BoolQ | PIQA | HellaSwag | WinoGrande | ARC-e | ARC-c | Avg. |
> | --------------          | -------------- | -------------- | -------------- | -------------- | -------------- | --------------| -------------- | -------------- | -------------- |
> | LLaMa-7B Full-precision       | 16/16/16 |  -  | 75.1 | 78.7 | 56.9 | 69.9 | 75.3 | 41.9 | 66.3 |
> | MinMax (E2M1)       | 4/4/4 | 32  | 47.3 | 53.1 | 25.7 | 50.7 | 25.1 | 22.4 | 37.4 |
> | SmoothQuant         | 16/4/4 |  512  | 54.1 | 62.8 | 41.5 | 52.6 | 50.6 | 32.9 | 49.1 |
> | FPT Baseline      | 4/4/4 |  32  | 57.3 | 56.6 | 30.2 | 51.1 | 37.71 | 23.2 | 42.7 |
> | FPT         | 4/4/4 |  32  | 64.2 | 73.5 | 47.8 | 63.7 | 65.9 | 33.6 | $\textbf{58.1}$ |
>
>
> **[Weakness 2:  More figures for better understanding]**
>
> Thanks for the suggestion. We added a visual representation to illustrate the distinction between FP and INT quantization in the preliminary section. Additionally, we moved Figure 2 one page ahead and provided more explanations in the figure caption to explain the process and make it easier to follow.

---

### Official Review · Reviewer_KRMt · 2023-08-08

**Soundness:** 4

**Excitement:**

3: Ambivalent: It has merits (e.g., it reports state-of-the-art results, the idea is nice), but there are key weaknesses (e.g., it describes incremental work), and it can significantly benefit from another round of revision. However, I won't object to accepting it if my co-reviewers champion it.

**Paper Topic And Main Contributions:**

This paper presents a 4-bit floating-point post-training quantization method for weights, activations, and embeddings in transformer models, including both Bert model and GPT models. The paper first proposes a Hessian-based joint search for determining the optimal exponent bias which consistently delivers desirable quantization results; then absorb the bias into the exponent bias of the floating point values. This technique is used to deal with high inter-channel variance in the transformer. Experimental results demonstrate that the proposed method yields the first usable FP4 quantized Bert model with competitive performance.

**Questions For The Authors:**

(1) Can you provide a comparison of inference latency between the proposed method and other methods (latency from hardware simulation is also acceptable)?
(2) How does the proposed method deal with the layernorm layer and residual addition in the transformer? Should fp32 or fp16 be used in these two cases?

**Reasons To Accept:**

(1) The proposed Hessian-base joint search method for determining the optimal exponent bias and maximal quantization value outperforms existing techniques in terms of stability and performance, establishing a strong baseline for floating point post-training quantization.
(2) The proposed pre-shifted exponent bias method can effectively addresses the challenge of high inter-channel variance in the transformer while incurring negligible computation cost.
(3) By combining the above two methods, the paper quantize both weights and activations in the Bert model to only 4-bit and achieves a significantly increased average GLUE score. The advantage is most obvious when activaiton bitwidth go below 8-bit.

**Reasons To Reject:**

(1) The inference latency comparison between the proposed method and other methods is lacking in the paper, making it difficult to evaluate the feasibility and comprehensive cost performance of the proposed method.
(2) It is necessary to supplement some experiments on the LLM generative tasks to verify the effectiveness of the proposed method. Since the method is PTQ, so dealing with models larger than LLaMa-7B should be possible and not that effort-taking?

**Reproducibility:**

3: Could reproduce the results with some difficulty. The settings of parameters are underspecified or subjectively determined; the training/evaluation data are not widely available.

**Reviewer Confidence:**

4: Quite sure. I tried to check the important points carefully. It's unlikely, though conceivable, that I missed something that should affect my ratings.

**Typos Grammar Style And Presentation Improvements:**

In contribution, "per-shifted" should be "pre-shifted". This caused quite some confusion at first.

---

> ### Author Rebuttal · Authors · 2023-08-28
>
> **[Weakness 1 and Question 1: Lack of inference latency comparison between the proposed method and other methods]**
>
> Comparing the inference latency between the proposed FP format and the INT format is highly challenging due to the sensitivity of the results to various hardware design choices. In cases where the bottleneck is the data transfer bandwidth, the latency difference resulting from INT8 and FP8 formats may be considered negligible or similar [1,2]. To avoid the complexities associated with hardware choices, in Sec. 5.4 of the paper, we conducted hardware-agnostic experiments focusing on the utilization of low-bit INT, FP, and mixed-format FP multiplication operators, such as adders, multipliers, and multiply-accumulate (MAC) units, in terms of hardware area.
>
> Our simulations reveal that FP formats generally incur higher costs for addition compared to the INT format, but they require less energy for multiplication. Furthermore, as the bit-width decreases, the disparity between INT and FP formats gradually diminishes, suggesting a reduction in latency difference as well. Additionally, FP8 has gained wide adoption across various hardware platforms and has shown no significant latency drawbacks compared to its INT counterpart. Consequently, we believe that the FPT approach not only serves as a robust baseline for future FP quantization research but also provides valuable insights for hardware researchers aiming to design ultra-low-bit transformer accelerators.
>
>
> **[Weakness 2: Experiments on the LLM generative tasks; More experiments on models larger than LLaMa-7B ]**
>
> We conducted experiments for generative tasks on LLaMa-7B, using perplexity evaluation on Wiki2 and C4, as shown in the following table. The results indicate that our proposed FPT framework consistently achieves better results than all the baselines, including MinMax and SmoothQuant.
>
> | Quant Methods          | #Bits (E/W/A) |  #Calib | Wiki2 | C4 |
> | ------          | ------  | ------  |------  | ------ |
> | LLaMa-7B Full-precision | 16/16/16 | - | 5.67 | 7.57 |
> | FP MinMax (E2M1)          | 4/4/4 |  32  |  100176  |  109849 |
> | SmoothQuant          | 16/4/4 |  512  | 174.87  | 107.73 |
> | FPT          | 4/4/4 |   32  |  $\textbf{8.66}$  | $\textbf{11.28}$ |
>
>
> Following your suggestion, we performed additional experiments on LLaMa-13B. The results indicate that both MinMax and SmoothQuant are unable to maintain accuracy in ultra-low bit settings. In contrast, the FPT method achieves an average accuracy of 63.1 with a mere 5.8 point loss in accuracy.
>
> | Quant Methods          | #Bits (E/W/A) |  #Calib | BoolQ | PIQA | HellaSwag | WinoGrande | ARC-e | ARC-c | Avg. |
> | --------------          | -------------- | -------------- | -------------- | -------------- | -------------- | --------------| -------------- | -------------- | -------------- |
> | LLaMa-13B Full-precision       | 16/16/16 |  -  | 77.9 | 79.2 | 59.9 | 72.6 | 77.4 | 46.4 | 68.9 |
> | FP MinMax (E2M1)       | 4/4/4 | 32  | 45.8 | 51.7 | 25.5 | 49.5 | 25.0 | 22.8 | 36.7 |
> | SmoothQuant         | 16/4/4 |  512  | 57.6 | 61.3 | 56.0 | 52.6 | 49.9 | 25.1 | 50.4 |
> | FPT         | 4/4/4 |  32 | 71.9 | 74.8 | 53.3 | 66.7 | 71.7 | 39.9 | $\textbf{63.1}$ |
>
> We will add these results in the experiment section of the paper to make it more comprehensive.
>
> **[Question 2: How does the proposed method deal with the layernorm layer and residual addition in the transformer? Should fp32 or fp16 be used in these two cases? ]**
>
> We follow previous works [3,4,5,6] which keep the residual addition and layernorm in fp32.
>
> **[Typos: In contribution, "per-shifted" should be "pre-shifted". This caused quite some confusion at first.
> ]**
>
> Thanks for the comments; we have thoroughly checked the paper and fixed typos in the manuscript.
>
>
> [1]: FP8 Quantization: The Power of the Exponent, Andrey Kuzmin, et al., 2022
>
> [2]: FP8 versus INT8 for efficient deep learning inference, Mart van Baalen, et al., 2023
>
> [3]: SmoothQuant: Accurate and Efficient Post-Training Quantization for Large Language Models, Guangxuan Xiao, et al., 2023
>
> [4]: Towards Efficient Post-training Quantization of Pre-trained Language Models, Haoli Bai, et al., 2022
>
> [5]: Brecq: Pushing the limit of post-training quantization by block reconstruction, Yuhang Li, et al., 2021
>
> [6]: QDrop: Randomly dropping quantization for extremely low-bit post-training quantization, Xiuying Wei, et al., 2022

---

### Official Review · Reviewer_SAuf · 2023-08-11

**Typos Grammar Style And Presentation Improvements:** 1. In Eq. 14, it should be J(X) not J(W)
**Soundness:** 2

**Excitement:**

3: Ambivalent: It has merits (e.g., it reports state-of-the-art results, the idea is nice), but there are key weaknesses (e.g., it describes incremental work), and it can significantly benefit from another round of revision. However, I won't object to accepting it if my co-reviewers champion it.

**Paper Topic And Main Contributions:**

This paper proposes to find optimal exponent bits and scaling factor in floating point quantization by using Hessian based loss function. It uses brute force search for find the optimal format, and it utilizes the absorbed bias for generating per channel variant used in multiplication. Experimental resutls are reported on 8bits and 6 bits Bert model and recent LLaMA model. Generally, the moder performance is well maintained after quantization.

**Questions For The Authors:**

1. how many bits for representing tilde b in Eq. 7, and for scaling factors like tilde alpha?
2. what is r_x in Algorithm 1, line 10? is it a set containing tilde b in Eq. 7, tilde alpha and v in Eq. 8? if so, what is the searching complexity? 100^3?
3. where does Eq.9 come from? is there a reference to understand it?
4. why does Eq. 10 only contain tilde alpha? what about v as in Eq. 8?
5. line 356 to line 358, why scaling factor cannot be a shared constant along the multiplication direction? it looks like in Eq. 10, each column of W has a shared scaling factor. why can't activations X have per channel scaling factor? it looks the section 4.3 is proposing a way of computing a shared scaling factor for each row of X.
6. line 399-401, it looks like you compute and store activation bias from calibration data, but how does the bias become zero in Eq. 20? if it is absorbed like in line 209, then it should not be tilde rho in Eq. 20?
7. line 422, is the searching of rho also for minimizing Eq. 15?
8. can you give an example of quantization result? for example in your 6 bit quantization result, how many of exponential bits, how many of mantissa bits?
9. according to Table 4, it looks like the algorithm variance can be huge, thus, it is reasonable to believe your Table 1 results may not be significantly better than others.
10. In Table 2, smoothquant seems reporting only quantization in 8 bits, did you run their algorithm on 4 bit quantization? how is your 8 bit quantization result compared with their model?

**Reasons To Accept:**

reasonable algorithm, good results

**Reasons To Reject:**

not good writing, hard to understand

**Reproducibility:**

2: Would be hard pressed to reproduce the results. The contribution depends on data that are simply not available outside the author's institution or consortium; not enough details are provided.

**Reviewer Confidence:**

4: Quite sure. I tried to check the important points carefully. It's unlikely, though conceivable, that I missed something that should affect my ratings.

---

> ### Author Rebuttal · Authors · 2023-08-28
>
> We appreciate the constructive feedback and the detailed comments. To clarify, our main contribution lies in being the first to successfully quantize a Transformer model into 4 bits with floating-point post-training quantization. In contrast,  previous studies primarily focused on 6-bit or 8-bit quantization methods.
>
> **[Q1.  how many bits for representing tilde b in Eq. 7, and for scaling factors like tilde alpha?]**
>
> $\tilde{b}$ and $\tilde{\alpha}$ are both FP numbers using 32-bit. As stated in lines 210 and 211 in the paper, $\tilde{b}$ denotes a relaxed tensor-wise real-valued exponent. $\tilde{\alpha}$ is also a real-valued number as indicated in line 209. “Real-valued” number is a term that has been widely used in the quantization literature [1], referring to the original numeric representation before quantization, which is FP32 in our context.
>
> **[Q2. what is r_x in Algorithm 1, line 10? is it a set containing tilde b in Eq. 7, tilde alpha and v in Eq. 8? if so, what is the searching complexity? 100^3?]**
>
> $r_x^i$ denotes one of the FP formats, for example, E1M2/E2M1 for FP4, as mentioned in line 1 of Algorithm 1. $r_x^i$ does not contain  $\tilde{b}$ / $\tilde{\alpha}$ / $v$, but each format corresponds to the associated $\tilde{b}$ / $\tilde{\alpha}$ / $v$. The search complexity varies with the bit-width. For example, for FP4, the search space should be 2 * 100, because we first choose a format from two choices (E1M2, E2M1) and then choose the $\tilde{b}$ / $\tilde{\alpha}$ / $v$ from 100 choices.
>
> **[Q3. where does Eq.9 come from? is there a reference to understand it?]**
>
> Eq.9 is a standard FP quantization formula, derived from Eq. 1. You may refer to Section 4.1 of [6] for more detail.  Sec.3.1 is a bit dense in formulas; we added a figure to make it easier to follow.
>
> **[Q4. why does Eq. 10 only contain tilde alpha? what about v as in Eq. 8?]**
>
> Eq. 10 only contains $\tilde{\alpha}$, as $v_x$ and $v_w$ are integrated into the $\tilde{X}$ and $\tilde{W}$, respectively. We will add $\tilde{X} = v \cdot \left\lfloor \frac{X_{\rm R}''}{\tilde{\alpha} \cdot v} \right\rceil$ after Eq. 10 to enhance the readability.
>
> **[Q5. line 356 to line 358, why scaling factor cannot be a shared constant along the multiplication direction? it looks like in Eq. 10, each column of W has a shared scaling factor. why can't activations X have per channel scaling factor? it looks the section 4.3 is proposing a way of computing a shared scaling factor for each row of X.]**
>
> The reason why activations X cannot have per-channel scaling factors is that the multiplication direction for the activation tensor operates on a per-row basis rather than per-channel, as visualized in Figure 5 in the appendix. Lines 356 to 358 indicate that to achieve efficient matrix multiplication, the scaling factor must be a shared constant along the multiplication direction. However, applying per-channel scaling factors for activations introduces challenges to efficient matrix multiplication, since the scaling factor is **NOT** a shared constant along the multiplication direction and cannot be extracted as shown in Equation 10. This limitation of per-channel scaling factors for activation X is also discussed in the SmoothQuant paper [2]. Therefore, in Sec. 4.3, we propose pre-shifted exponent bias which utilizes the characteristic of FP format to compute a shared scaling factor for each **channel**(column) of X while being able to maintain efficient matrix multiplication
>
> **[Q6. line 399-401, it looks like you compute and store activation bias from calibration data, but how does the bias become zero in Eq. 20? if it is absorbed like in line 209, then it should not be tilde rho in Eq. 20?]**
>
> Thank you for your question. We acknowledge that this particular part is complicated, and as a result, we have reorganized the explanations and equations as follows:
>
> Before the pre-shifted exponent bias,
>
> \begin{aligned}
> X_{\rm FP} &=  2^{-\tilde{\mathbf{b}_j}} \cdot(-1)^s2^{p-0}(1+\frac{d_1}{2}+...+\frac{d_m}{2^m}) \\
>     = 2^{-\tilde{\rho}}(-1)^s2^{p-\mathbf{b}^{ori}_j}(1+\frac{d_1}{2}+...+\frac{d_m}{2^m})
> \end{aligned}
>
>
> \begin{aligned}
> W_{\rm FP}=2^{-\tilde{\mathbf{b}}^{W}}(-1)^s2^{p-0}(1+\frac{d_1}{2}+...+\frac{d_m}{2^m})
> \end{aligned}
>
> After pre-shifted exponent bias,
>
> \begin{aligned}
> X_{\rm FP}&=2^{-\tilde{\rho}}(-1)^s2^{p-0}(1+\frac{d_1}{2}+...+\frac{d_m}{2^m})
> \end{aligned}
>
> \begin{aligned}
> W_{\rm FP}=2^{-\tilde{\mathbf{b}}^{W}}(-1)^s2^{p-\mathbf{b}^{ori}_j}(1+\frac{d_1}{2}+...+\frac{d_m}{2^m})
> \end{aligned}
>
> Looking at these formulas, we can see the proposed pre-shifted exponent bias transfers the activation bias to the weight tensor, as visualized in Figure. 2.
>
> **[Q7. line 422, is the searching of rho also for minimizing Eq. 15?]**
>
> Yes, we will add it in line 422, that the algorithm searches for the optimal $\tilde{\rho}_{\mathbf{X}}$ by minimizing Eq.15.
>
> **[Q8. Can you give an example of quantization result? for example in your 6 bit quantization result, how many of exponential bits, how many of mantissa bits?]**
>
> As mentioned in lines 304 and 305 of Section 4.2, the search for the optimal format is performed on a layer-by-layer basis. This implies that each weight and activation tensor in different layers may have different search results. We listed the search results of each linear layer in the 6-bit quantized Bert for your reference and will add these results in the Appendix of the paper. From the result, we can observe that the majority of modules reach E2M3 once the search is completed. However, certain modules, such as the activation of output.dense and intermediate.dense, require E3M2 to effectively handle distributions with longer tail.
>
>
> | Layer # | attn.query (W/A) | attn.key (W/A) | attn.value (W/A) | attn.output.dense (W/A) | intermediate.dense (W/A) | output.dense (W/A) |
> | ----- | ----- |----- | ----- | ----- | ----- | ----- |
> | 0 |  E2M3/E2M3 | E2M3/E2M3 | E2M3/E2M3 | E2M3/E2M3 | E2M3/E2M3 | E2M3/E3M2 |
> | 1 |  E2M3/E2M3 | E2M3/E2M3 | E2M3/E2M3 | E2M3/E2M3 | E2M3/E2M3 | E2M3/E3M2 |
> | 2 |  E2M3/E2M3 | E2M3/E2M3 | E2M3/E2M3 | E2M3/E2M3 | E2M3/E2M3 | E3M2/E3M2 |
> | 3 |  E2M3/E2M3 | E2M3/E2M3 | E2M3/E2M3 | E2M3/E2M3 | E2M3/E2M3 | E3M2/E3M2 |
> | 4 |  E2M3/E2M3 | E2M3/E2M3 | E2M3/E2M3 | E2M3/E3M2 | E2M3/E3M2 | E3M2/E3M2 |
> | 5 |  E2M3/E2M3 | E2M3/E3M2 | E2M3/E3M2 | E2M3/E2M3 | E2M3/E2M3 | E3M2/E3M2 |
> | 6 |  E2M3/E2M3 | E2M3/E2M3 | E2M3/E2M3 | E2M3/E2M3 | E2M3/E2M3 | E2M3/E3M2 |
> | 7 |  E2M3/E2M3 | E2M3/E2M3 | E2M3/E2M3 | E2M3/E2M3 | E2M3/E3M2 | E2M3/E3M2 |
> | 8 |  E2M3/E2M3 | E2M3/E2M3 | E2M3/E2M3 | E2M3/E2M3 | E2M3/E3M2 | E2M3/E3M2 |
> | 9 |  E2M3/E2M3 | E2M3/E2M3 | E2M3/E2M3 | E2M3/E2M3 | E2M3/E2M3 | E2M3/E3M2 |
> | 10 |  E2M3/E2M3 | E2M3/E2M3 | E2M3/E2M3 | E2M3/E2M3 | E2M3/E2M3 | E2M3/E3M2 |
> | 11 |  E2M3/E2M3 | E2M3/E2M3 | E2M3/E2M3 | E2M3/E2M3 | E2M3/E2M3 | E2M3/E3M2 |
>
> **[Q9. according to Table 4, it looks like the algorithm variance can be huge, thus, it is reasonable to believe your Table 1 results may not be significantly better than others.]**
>
> Table 4 is an ablation for different numbers of calibration sentences, not the variance of experiments. Too few calibration data will lead to accuracy degradation, for example, 32/64 sentences. Thus, we recommend using 128 data points, which delivers stable and satisfactory accuracy on most datasets. Note that other baseline methods [3, 4, 5], which employ a much larger calibration set with 4096 sentences. For the experiment variance, We also did multi-trial experiments on the Bert model, the variance is only 0.002 point.
>
> **[Q10. In Table 2, smoothquant seems reporting only quantization in 8 bits, did you run their algorithm on 4 bit quantization? how is your 8 bit quantization result compared with their model?]**
>
> Yes, we reimplemented Smoothquant with a 4-bit setting and reported the results.
>
> According to your suggestion, we compare the 8-bit quantization results with Smoothquant.  It is worth noting that naive MinMax FP8 quantization can already achieve nearly lossless accuracy, as also mentioned in the paper. Consequently, the difference in performance between MinMax, SmoothQuant, and our proposed FPT methods is insignificant in the 8-bit setting, as demonstrated below:
>
> | Quant Methods          | #Bits (E/W/A) |  #Calib | BoolQ | PIQA | HellaSwag | WinoGrande | ARC-e | ARC-c | Avg. |
> | --------------          | -------------- | -------------- | -------------- | -------------- | -------------- | --------------| -------------- | -------------- | -------------- |
> | LLaMa-7B Full-precision       | 16/16/16 |  -  | 75.1 | 78.7 | 56.9 | 69.9 | 75.3 | 41.9 | 66.3 |
> | MinMax (E4M3)       | 8/8/8 | 32  | 74.86  | 78.62 | 56.8 | 69.46 | 75.46 | 41.55 | 66.12 |
> | SmoothQuant         | 16/8/8 |  512  | 73.95 | 77.46 | 55.02 | 69.61 | 74.34 | 37.37 | 64.62 |
> | FPT         | 8/8/8 |  32  | 75.6 | 78.18 | 56.57 | 70.24 | 74.62 | 40.7 | 65.98 |
>
> **[Typo: In Eq. 14, it should be J(X) not J(W)]**
> Thanks for pointing this out, we will correct the typo in our final version of the paper.
>
> [1]: A Survey of Quantization Methods for Efficient Neural Network Inference, Amir Gholami, et al., 2021
>
> [2]: SmoothQuant: Accurate and Efficient Post-Training Quantization for Large Language Models, Guangxuan Xiao, et al., 2023
>
> [3]: Towards Efficient Post-training Quantization of Pre-trained Language Models, Haoli Bai, et al., 2022
>
> [4]: Brecq: Pushing the limit of post-training quantization by block reconstruction, Yuhang Li, et al., 2021
>
> [5]: QDrop: Randomly dropping quantization for extremely low-bit post-training quantization, Xiuying Wei, et al., 2022
>
> [6]: FP8 Quantization: The Power of the Exponent, Andrey Kuzmin, et al., 2022

---

### Official Review · Reviewer_v99B · 2023-08-12

**Soundness:** 3

**Excitement:**

3: Ambivalent: It has merits (e.g., it reports state-of-the-art results, the idea is nice), but there are key weaknesses (e.g., it describes incremental work), and it can significantly benefit from another round of revision. However, I won't object to accepting it if my co-reviewers champion it.

**Paper Topic And Main Contributions:**

* This paper has proposed a channel-wise floating quantization method for transformer-based models which has shown prominent performance in NLP and CV tasks.
* This work brought hessian-based loss into the FP quantization and analyze the inter-intra channel variance to show the motivation.

**Questions For The Authors:**

* I am not quite clear how is floating number being compressed to ultra-low bit such as 4bit, how does it maintain sufficient information? For example, with only 1 bit for mantissa, is it fair to use such low bit? However, as long as the result in paper is credible, it is really impressive then.




**Reasons To Accept:**

* Floating point quantization by replacing the scaling factor or bias from real value to exponent may help handle the outlier problem during quantization.

* Good performance showed with sufficient experiments.

* Detailed and clear explanation on the workflow with good expression.

**Reasons To Reject:**

* The Hessian-based objective subsection 4.1 is not new.

* As far as I know the mainstream DNN accelerator do not hold specific for such kind of quantization scheme. Just a bit skeptical on the practicality. You see the bit width itself is not the only concern as speed/computation cost matters more in most cases (although I would agree that pre-shifted bit bias can help with computation cost).


**Reproducibility:**

2: Would be hard pressed to reproduce the results. The contribution depends on data that are simply not available outside the author's institution or consortium; not enough details are provided.

**Reviewer Confidence:**

2: Willing to defend my evaluation, but it is fairly likely that I missed some details, didn't understand some central points, or can't be sure about the novelty of the work.

---

> ### Author Rebuttal · Authors · 2023-08-28
>
> **[Weakness 1: The hessian-base objective subsection 4.1 is not new]**
>
> Thanks for the comments. Prior studies that employed hessian-based metrics focused solely on INT PTQ. Different from INT quantization, FP quantization involves searching the format choices as well as the associated maximum quantization value, and the hessian-based search method taking these factors into consideration is absent in the literature. As a result, the proposed hessian-base method for FP PTQ establishes a strong baseline that can already achieve lossless results in both 8-bit and 6-bit configurations, making it a robust cornerstone for future endeavors.
>
> **[Weakness 2: Concerns regarding the practicality of the proposed methods on the current mainstream DNN accelerator]**
>
> The primary focus of this paper is to investigate the effectiveness of low-bit FP formats from a generally applicable perspective, without delving into the intricate details of hardware implementation for dedicated DNN accelerators. We believe that such an assessment would be highly sensitive to specific hardware design choices, considering the multitude of factors involved, such as overflow handling, NaN management, multiplier and accumulator design, and support for sparsity, among others [1,2,3]. Nevertheless, we did analyze the hardware utilization of low-bit INT, FP, and mixed-format FP multiplication operators in Section 5.4, which offers a more hardware-agnostic perspective. Our findings indicate that FP formats generally incur higher costs for addition compared to the INT format but require less energy for multiplication. Moreover, the discrepancy between the INT and FP formats gradually diminishes as the bit-width decreases, suggesting a reduction in latency difference between INT and FP as well.
>
> Furthermore, since FP8 is already widely supported by various hardware platforms, we believe that adopting similar hardware implementations to accommodate FP formats with bit-widths lower than 8 bits should be relatively straightforward. Our proposed FPT framework can serve as a valuable reference for those seeking to develop their own low-bit FP transformer accelerators based on their own hardware use case.
>
> **[Question 1:  how is floating number being compressed to ultra-low bit such as 4bit]**
>
> Thanks for this insightful question. We also note that the difference between ultra-low FP numbers and INT numbers is small. For example, compared to INT4, FP4 has only one more representation which is E2M1, since E1M2 is identical to INT4. As a result, we can observe that in Table 1, the naive MinMax FP did not achieve better results compared to integer quantization. Nevertheless, the proposed pre-shifted exponent bias utilizes the characteristics of the FP format to handle the high channel-wise variance problem and improve the FP quantization accuracy significantly especially for FP4 setting. The code will be released for future works to easily follow up and reproduce our results.
>
> [1]: FP8 formats for deep learning, Paulius Micikevicius, et al., 2022
>
> [2]: FP8 Quantization: The Power of the Exponent, Andrey Kuzmin, et al., 2022
>
> [3]: FP8 versus INT8 for efficient deep learning inference, Mart van Baalen, et al., 2023

---

### Meta-Review · Area_Chair_bj57 · 2023-09-16

**Recommendation:** 5

**Metareview:**

This paper presents a floating-point-based post-training quantization regime for the transformer models. Reviewers initially had some problems with the clarity of presentation and the novelty of the methods, the authors have addressed the concerns. Reviewers unanimously agreed to accept this paper, thus, the AC deemed this paper should be accepted to the main conference.

---

### Decision · Program_Chairs · 2023-10-07

**Decision:**

Accept-Main

**Comment:**

This paper presents a floating-point-based post-training quantization regime for the transformer models. Reviewers initially had some problems with the clarity of presentation and the novelty of the methods, the authors have addressed the concerns. Reviewers unanimously agreed to accept this paper, thus, the AC deemed this paper should be accepted to the main conference.